# Principles and Current Clinical Landscape of Multispecific Antibodies against Cancer

**DOI:** 10.3390/ijms22115632

**Published:** 2021-05-26

**Authors:** Mariam Elshiaty, Hannah Schindler, Petros Christopoulos

**Affiliations:** 1Thoraxklinik and National Center for Tumor Diseases (NCT) at Heidelberg University Hospital, 69126 Heidelberg, Germany; mariam.elshiaty@med.uni-heidelberg.de (M.E.); hannahlena.schindler@med.uni-heidelberg.de (H.S.); 2Translational Lung Cancer Center Heidelberg, Member of the German Center for Lung Research (DZL), 69126 Heidelberg, Germany

**Keywords:** bispecific antibodies, multispecific antibodies, monoclonal antibodies, therapeutic antibodies, antibody engineering

## Abstract

Building upon the resounding therapeutic success of monoclonal antibodies, and supported by accelerating progress in engineering methods, the field of multispecific therapeutic antibodies is growing rapidly. Over 140 different molecules are currently in clinical testing, with excellent results in recent phase 1–3 clinical trials for several of them. Multivalent bispecific IgG-modified formats predominate today, with a clear tendency for more target antigens and further increased valency in newer constructs. The strategies to augment anticancer efficacy are currently equally divided between disruption of multiple surface antigens, and additional redirection of cytotoxic T or NK lymphocytes against the tumor. Both effects complement other modern modalities, such as tyrosine kinase inhibitors and adoptive cell therapies, with which multispecifics are increasingly applied in combination or merged, for example, in the form of antibody producing CAR-T cells and oncolytics. While mainly focused on B-cell malignancies early on, the contemporary multispecific antibody sector accommodates twice as many trials against solid compared to hematologic cancers. An exciting emerging prospect is the targeting of intracellular neoantigens using T-cell receptor (TCR) fusion proteins or TCR-mimic antibody fragments. Considering the fact that introduction of PD-(L)1 inhibitors only a few years ago has already facilitated 5-year survival rates of 30–50% for per se highly lethal neoplasms, such as metastatic melanoma and non-small-cell lung carcinoma, the upcoming enforcement of current treatments with “next-generation” immunotherapeutics, offers a justified hope for the cure of some advanced cancers in the near future.

## 1. Introduction

The tortuous, 2.5 billion-years-long path from immunoglobulin (Ig)-like domains of archaeal flagellins to the antibodies (Ab) of jawed vertebrates is one of the most intriguing discoveries in evolutionary biology [1,2,3]. Equally impressive are the accomplishments of modern genetic engineering, whose further variation of basic Ig building blocks could produce over 40 different molecular formats of therapeutic antibodies during the last two decades [4]. As times change, priorities shift, and pathogen defense has today largely been succeeded by a much more challenging task: the fight against cancer [5].

The concept that antibodies could be used as “magic bullets” against human maladies dates back to their discovery in the late 19th century [6] and the gradual recognition that they can bind a virtually unlimited number of antigens with a high specificity and affinity [7]. However, it was not until discovery of the “hybridoma” technology in 1975 [8], complemented by various humanization techniques a few years later [9], that scientists managed to harness this power: monoclonal antibodies could now be produced in large quantities after injecting a mouse (later, rat or other mammal) with the desired antigen, and fusing the respective splenocytes with suitable myeloma cell lines [10]. In the next step, two hybridomas were fused (“hybrid hybridoma”, aka “quadroma”) to produce bispecific antibodies without the protein denaturation steps necessary for chemical cross-linking, which could potentially adversely affect binding properties [11,12].

Improved function is the main incentive behind development of multispecific constructs. For every antibody, the “classic” mode of action falls into two broad categories: (i) “disruptive” with respect to the epitope-bearing target molecule, i.e., blocking or activating signals, neutralizing antigens, or causing internalization and degradation of surface receptors and (ii) “recruiting”, i.e., activating immune cells and/or other effector molecules, like the complement [13]. OKT3 (aka “muromonab-CD3”), for example, the first monoclonal antibody to ever achieve regulatory approval in 1986, is a typical product of the first category, used to suppress T-cell function in patients with glucocorticoid-resistant rejection of allogeneic renal, heart and liver transplants [14]. Rituximab, on the other hand, a CD20-specific monoclonal antibody still widely used since its approval in 1997, kills B- cells by combining signaling-induced death with cellular and complement-mediated cytotoxicity [15]. Compared to monospecific monoclonal antibodies, multispecific constructs potentiate antibody-mediated effects, for example, they can potentially “disrupt” multiple instead of one tumor-associated antigens (TAA) owing to more antigen-binding regions, and/or they can “recruit” and activate immune cells even stronger, since they use dedicated antigen-binding sites for this. The functional augmentation facilitated by multispecificity is clinically relevant: it translates into improved response rates, for example, approximately 50% with the newer anti-CD20/CD3 bispecific antibodies as monotherapy in B-cell non-Hodgkin’s lymphomas (B-NHL) which do not respond to rituximab any more [16], can delay development of resistance, and simplifies drug development compared to the more complicated, expensive and time-consuming procedures necessary for launching of multiple monospecific products instead [4].

Wide adoption of genetic engineering facilitates today’s exploitation of the huge potential inherent in multispecific antibodies: suitable polypeptide chains are designed in silico and expressed in various host systems, most frequently CHO cells and *E. coli*, followed by purification and assembly of the various components in vitro [17,18,19]. Appropriate antigen-binding properties, high yield, high thermal and chemical stability, good solubility, and low viscosity have key importance for large-scale production and clinical applicability [20,21]. The biochemical basis of these characteristics and our ability to manipulate them lie rooted in the modular antibody structure (Figure 1a).

## 2. Antibody Structure and Approaches to Multispecificity

The typical structure of human antibodies is represented by the IgG isotype, which is the most prevalent class [22]. X-ray crystallographic and electron-microscopic studies have revealed this to be a heterotetrametric, roughly Y-shaped protein with axial symmetry, which consists of two identical heavy (“H”, approximately 50 kDa each), and two identical “light” (“L”, approximately 25 kDa each) polypeptide chains, linked together by disulfide bonds (Figure 1a) [23,24,25]. Basic building block for both chain types is the “Ig domain”, aka “Ig fold”, which is a sandwich-like structure formed by two sheets of 7–9 antiparallel β-strands arranged with a Greek-key topology [26]. This basic motif appears to be conserved throughout the evolution of life, presumably because its efficient and compact folding provides a suitable substrate for numerous essential recognition, binding and adhesion processes carried out by members of the large Ig protein superfamily [1]. Each human antibody heavy chain consists of four domains, three constant ones (termed C_H_1, C_H_2 and C_H_3) and one variable (V_H_), while the light chains consist of one constant (C_L_) and one variable domain (V_L_) each. Limited digestion with the cysteine protease papain splits the IgG antibody in three equal-sized portions, namely two antigen-binding fragments (Fab), each consisting of one light chain bound with the V_H_ and C_H_1 domains of its partner heavy chain, and one crystallizable fragment (Fc), which contains the remaining constant domains of the heavy chain (C_H_2 and C_H_3, Figure 1a) [27]. Within the Fab region, the side-by-side arrangement of the V_L_ and V_H_ domains brings discrete amino acid loops between their β-sheets (“complementarity-determining regions”, CDR) together to form the antigen-binding site at the outer tip (Figure 1a). Direct linking of V_L_ and V_H_ by a peptide chain creates the single-chain variable fragment (scFv), an artificial construct that can also effectively bind antigens (Figure 1b) [28]. Interestingly, the antigen specificity of most antibodies shows a predominant dependence on the CDRs contributed by mainly the V_H_ rather than the V_L_ domain, which takes an extreme form in the heavy chain-only antibodies naturally produced by camelids and sharks [29], and is exploited by “single-domain antibodies” (sdAb), also referred to as nanobodies, consisting of a single VHH (variable heavy chain homodimer) thereof (Figure 1b) [30].

Due to its axial symmetry, the IgG can bind two identical epitopes with two binding sites (“paratopes”), one on each Fab arm, i.e., it is “monospecific”, but “bivalent” [31]. Generally, the valency of an antibody refers to the total number of epitopes that it can bind, and its specificity to the number of different structures among them. Naturally occurring antibodies are generally monospecific, which allows them to cross-link large numbers of antigen molecules and thus amplify the immune response against them [22]. Nonetheless, for IgG4 and possibly also other Ig subclasses, reducing conditions in the blood or cell surfaces can break up disulfide bonds and facilitate Fab-arm exchange (FAE) which gives chimeric bispecific antibodies with anti-inflammatory activity (Figure 1c) [32,33]. Generation of the first artificial bispecific antibodies in 1961 mimicked this naturally occurring FAE: by reducing and reoxidizing the F(ab’)^2^ fragments derived by peptic digestion of two different antibodies, their univalent Fab were recombined into new F(ab’)^2^ molecules that could precipitate a mixture of their cognate antigens, but neither of them in pure form [34,35]. After 1975, “quadromas” became a relatively easy method to generate full IgG-like bispecific molecules (Figure 1c), but the low yield of the desired antibody (12.5% with random pairing of heavy and light chains), together with the difficulty to isolate it from the closely related mispaired contaminants, remained a significant problem [36]. Over the following decades, many different methods within the constraints of the classical “IgG-like” format were devised to overcome or circumvent the “mispairing” problem of heavy and light chains produced by quadromas or genetic engineering (Figure 1c): rat/mouse quadromas, which took advantage of the species-restricted heavy/light chain pairing and the differential affinity of protein A for mouse and rat heavy chains (“TrioMabs”) [37,38]; various “knobs-into-holes” (KiH) or other techniques of inducing complementary mutations in the sequences of heavy and/or light chains in order to force the desired heterodimerization [39,40,41,42] or facilitate controlled FAE (“Duobodies”) [43]; “common-light-chain” antibodies, in which the two distinct paratopes on each Fab arm utilize the same light chain paired with a different heavy chain in order to bind its target antigen [44]; “κ-λ” bodies, which utilize the same heavy, but different light chains for the two paratopes, in order to obviate the need for artificial mutations or linkers that may result in poor stability and increased potential immunogenicity [45]; “CrossMabs”, with swapping of either the variable or the constant domains between light and heavy chains to create two asymmetric Fab arms that force the desired light chain pairing, while preserving the binding properties of the respective paratope [46,47]; electrostatic steering effects [48,49]; IgG/A chimeras, aka “strand exchange engineered domain bodies” (“SEEDBodies”) [50]; the proprietary “Azymetric” heterodimeric Fc [51]; “dual action Fab” (DAF, aka “two-in-one” antibodies), which use the same heavy and light chains to recognize two unrelated antigens via differential use of their two paratopes [52,53]; “DutaMabs” or “DutaFabs”, in which each Fab arm contains two different paratopes, each utilizing only 3 out of the 6 available CDRs [54].

“IgG-modified” formats (Figure 1d) provide additional solutions to this problem, e.g., in “dual-variable-domain” (DVD) antibodies, each chain contains two variable domains, so that bispecificity is ensured irrespective of light chain pairing [55]. Moreover, using genetic engineering, the antigen-binding moieties Fab, Fv and VHH can be combined freely with each other, or linked to IgGs, resulting in huge structural variability (Figure 1d,e and Table 1) [4,56]. In addition, monospecific and bispecific formats can be combined in order to increase specificity, valency, or both; for example a highly active tetravalent and tetraspecific “four-in-one” antibody against EGFR, HER2, HER3 and VEGF was generated by combining the DVD, CrossMab and KiH technologies [57]. It should also be noted that the functionality of multispecifics can further be extended through fusion with other non-Ig proteins; for example, the T-cell receptor (TCR) [58], the TGFβ receptor [59], an IL-15 moiety in case of Trispecific Killer Cell Engagers (TriKEs) [60], various payloads [61], and by conjugation with Engeneic Delivery Vehicles (EDV), i.e., bacterially-derived nanocells coated with bispecific antibodies for the targeted delivery of cytotoxics, siRNA and other cargo to the tumor cells (Figure 1f) [62,63].

Among the numerous possible variations, the single most important structural characteristic of multispecific antibodies is whether they contain an Fc region or not. Fc-based constructs are generally larger, have longer half-lives (typically a few weeks) due to recycling by the neonatal Fc receptor (FcRn) and glomerular preservation [64,65], show improved solubility and stability, trigger cytotoxic [66] and T-cell priming effects [67], and can be purified using established affinity chromatography workflows [68]. In contrast, Fc-free, antibody fragment-based constructs are usually rapidly cleared from the blood (within minutes), which necessitates either administration by continuous infusion, or fusion with carrier moieties, such as human serum albumin (HSA) or polyethylene glycol (PEG), in order to extend their half-lives [69,70,71]. At the same time, specific advantages of smaller multispecifics can be better diffusion into the tumor tissue, higher potency due to closer proximity of interactions in the two paratopes, ease of large-scale production in microbial systems, and less immune-related adverse effects due to lack of Fc [72,73,74].

In terms of functionality, the basic distinction is between the “classic” and the “recruiting” or “redirecting” mode of antibody action, in which at least one binding site engages invariable immune cell receptors, for example, CD3 on T, or CD16 on NK cells. The combined structural (“Fc-based” vs. “Fc-free”) and functional (“classic” vs. “recruiting”) characterization is a useful framework to contextualize multispecific constructs (Table 1).

## 3. Bispecific Antibodies

The resounding therapeutic success of monoclonal antibodies, such as rituximab, trastuzumab, cetuximab and bevacizumab, the preclinical superiority of multispecific constructs, and technical advances in antibody engineering have facilitated rapid growth in the field during the last decade [75]. While only two bispecific antibodies are approved by the U.S.A. Food and Drug Administration (FDA) and the European Medicines Agency (EMA) at present, blinatumomab (Amgen) and emicizumab (Roche), more than 120 candidates are in clinical testing, and the market opportunity of multispecifics is estimated to exceed USD 18 billion until 2028 [76]. Autoimmune diseases (arthritis, asthma, diabetes), Alzheimer’s, infections (pneumonia) and hemophilia combined are easily dwarfed by oncology, which is the intended field for approximately two-thirds of these drugs, and builds the main scope of this review [77].

Clinical trials were identified by searching the ClinicalTrials.gov database on 26 April 2021 using the keywords “bispecific antibody”, “trispecific” and “oncology”, followed by manual verification of results and additional search of pharmaceutical companies’ pipelines. Overall, 324 trials were identified, using 146 different multispecific antibodies in 40 different molecular formats (Table 1, Figure 2 and Appendix A). Constructs with a “classical” and “redirecting” mode of action were balanced (156 vs. 166), while multispecific and multivalent formats increased over time (Figure 2a). Trials in solid tumors (*n* = 223) outnumbered trials in hematologic malignancies (*n* = 101) by a ratio of 2:1.

### 3.1. IgG-Like Antibodies

The first bispecific antibody approved by the EMA in oncology was the chimeric TrioMab (Figure 1c, Table 1) catumaxomab (Removab) in 2009, half anti-CD3 rat IgG2b, half anti-EpCAM mouse IgG2a, for intraperitoneal treatment of patients with malignant ascites (Figure 1c, Table 1) [78]. In a pivotal clinical study of 258 patients with recurrent malignant ascites (NCT00836654), catumaxomab could significantly prolong the median puncture-free survival to 52 days for ovarian and 37 days for other cancers vs. 11 and 14 days in controls, respectively [79]. In 2017, it was withdrawn from the European market for commercial reasons [80], but will probably soon return, as the Chinese biopharmaceutical company Lintonpharm recently announced a phase 3 trial of catumaxomab for patients with advanced gastric cancer with peritoneal carcinomatosis in China (NCT04222114), while phase 1 and 2 clinical trials in patients with non-muscle-invasive bladder cancer are planned both in China and in Germany (NCT04799847, NCT04819399). Notably, several other similar TrioMab antibodies developed by Trion Pharma and Fresenius according to the same principle, e.g., the HER2-directed etumaxomab, and the CD20-directed FBTA05, failed [81,82].

Today, the most common principle behind “two-halves” bispecific antibodies in clinical trials is the controlled FAE (Figure 1c, Table 1, Figure 2). One very efficient method is the proprietary DuoBody platform of Genmab in collaboration with Janssen, which utilizes matched and destabilizing mutations in CH3 to increase FAE yield >95% and was used to generate the bispecific, Fc-silenced anti-EGFR/cMet antibody Amivantamab (JNJ-61186372) [43]. Amivantamab‘s mode of action combines EGFR/MET downmodulation and NK/macrophage-dependent cancer-cell killing with minimal toxicity [83,84]. Based on very promising results of the Chrysalis phase 1/2 study for non-small-cell lung cancer (NSCLC) with *EGFR* Exon20 insertions (Ex20^+^, NCT02609776), an application for accelerated approval was filed by the FDA and EMA in late 2020, while the drug is already available through an international compassionate use program for these patients [85]. In addition, amivantamab is currently being tested in combination with platinum/pemetrexed as first-line therapy for *EGFR* Ex20^+^ NSCLC in the phase 3 PAPILLON trial (NCT4538664), in combination with the third-generation EGFR tyrosine kinase inhibitor (TKI) lazertinib as first-line therapy for NSCLC with *EGFR* Ex19 deletions or p.L858R mutations in the phase 3 MARIPOSA study (NCT04487080) [86], and for subcutaneous administration (NCT04606381). Another DuoBody is GEN3009, which targets two different paratopes of the B-cell antigen CD37 [87], shows hexamerization and enhanced complement-dependent cytotoxicity (CDC) due to an artificial E430G mutation in its Fc region (“DuoHexaBody”) [88], and is currently in phase 1 testing for r/r B-NHL (NCT04358458).

Several further Duobodies are T-cell redirecting. Epcoritamab (GEN3013), targeting CD20/CD3, is administered subcutaneously [89] and is currently in phase 3 testing for r/r diffuse large B-cell lymphoma (DLBCL, NCT04628494) after inducing remissions in the majority (67–100%) of r/r B-NHL in a previous phase 1/2 trial (NCT03625037), including patients failing CAR-T cells [90]. Adverse effects are well-manageable, the most common being fever, local injection site reactions and fatigue, without any incidence of grade 3-4 cytokine-release syndrome (CRS) [90]. Teclistamab (JNJ-64007957) is an anti-CD3/B-cell maturation antigen (BCMA) Duobody currently in phase 2 testing in r/r multiple myeloma (MM) (NCT04557098) [91,92], as is also Talquetamab (JNJ-64407564), an anti-GPRC5D/CD3 DuoBody (NCT04634552). In the myeloid space, the anti-CD3/CD33 Duobody JNJ-67571244 is in phase 1 testing for r/r acute myeloid leukemia (AML) and high-risk myelodysplastic syndromes (MDS) (NCT03915379) [93,94]. A slightly different mode of action is employed by the anti-PD-L1/4-1BB DuoBody GEN1046, which activates T-cells and NK cells by simultaneously blocking PD-L1 on tumor tissue and triggering the co-stimulatory checkpoint 4-1BB [95]. Across various r/r solid malignant tumors, it showed a disease control rate of 66% (40/61) in an ongoing phase 1/2 trial (NCT03917381), while it has hepatitis, hypothyroidism, and fatigue as the main adverse events [96]. “Two-half” antibodies have also been developed using other platforms, for example, the biparatopic anti-HER2 antibody MBS301 by Mabworks currently in phase 1 (NCT03842085), the anti- BCMA/CD3 Elranatamab (PF-06863135) by Pfizer in phase 2 for r/r MM (NCT04649359), the anti-PD1/PD-L1 IBI318 by Innovent in several phase 1/2 trials for advanced tumors, alone or combined with lenvatinib or chemotherapy (see Appendix A), and the anti-PD1/PD-L1 LY3434172 based on Zymeworks’ Azymetric platform in phase I testing also for advanced cancers (NCT03936959) [97].

Another “two-halves” IgG-like format frequently utilized in contemporary clinical trials are bispecific antibodies with common light chains (CLC, Figure 1c, Table 1), which, like TrioMabs, require multiple purification steps in order to isolate the desired bispecific antibody from the 2 contaminating monospecific antibodies [98]. Navicixizumab (OMP-305B83) which targets the Notch/Delta-like ligand 4 (DLL4) and the vascular endothelial growth factor (VEGF), showed a disease control rate (DCR) of 32% as monotherapy in a phase 1a trial (NCT02298387) of various solid tumors, and the phase 1b results in combination with chemotherapy are pending (NCT03030287) [99,100]. Odronextamab (REGN1979) is an anti-CD3/CD20 bispecific antibody with comparable efficacy as Epcoritamab in r/r B-NHL, and also very good tolerability (NCT02290951) [101], for which a phase 2 trial is currently ongoing (NCT03888105). Regeneron is also developing REGN5458 and REGN5459, both anti-BCMA/CD3 tested for multiple myeloma (NCT03761108, NCT04083534), and several other CLC antibodies for various solid tumors (please see Appendix A). The CLC technology is also used by Merus with the anti-HER2/HER3 Zenocutuzumab (MCLA-128) in phase 1/2 testing for tumors with NRG1 fusions (NCT02912949) as flagship product, by Alphamab with the biparatopic anti-HER2 KN026 in phase 1/2 testing for breast and other HER2^+^ solid cancers (7 active studies, listed in the Appendix A), and by Chugai Pharmaceutical with the GPC3/CD3 ERY974, whose target Glypican3 is a membrane-bound heparan sulfate proteoglycan expressed in 70–80% of hepatocellular carcinomas and various other human cancers [102] (NCT02748837, please see Appendix A).

To simplify the purification steps necessary for CLC, but still benefit from the IgG-like format, κλ-bodies (Figure 1c, Table 1) were developed, which have common heavy, rather than light chains. In this case, two monospecific and one bispecific antibody are also produced, but the purification follows three simple steps using affinity resins binding to (1) constant regions of the heavy chains, then (2) to the constant regions of the κ, and (3) λ chains [45]. TG-1801 (of TG Therapeutics, formerly NI-1701 by Novimmune), an anti-CD47/CD19 antibody blocking the CD47 checkpoint on CD19-positive cells to induce antibody-dependent cellular phagocytosis (ADCP) and cell-mediated cytotoxicity (ADCC), is currently in phase 1 testing for r/r B-NHL (NCT03804996, NCT04806035) [103], while other κλ bodies are still in preclinical development.

The CrossMab platform (CrossMab 1:1, Figure 1c, Table 1) by Roche reduces mispaired contaminants by swapping domains between the light and heavy chains, and has generated several promising antibodies. Most advanced is the anti-CD20/CD3 Mosunetuzumab (RG7828), which is currently in phase 3 testing for r/r B-NHL (NCT04712097) [104]. Moreover, the anti-PD-1/TIM-3 bispecific RO7121661 is being currently evaluated in a phase 1 trial (NCT03708328) of various solid tumors, however, Vanucizumab, an anti-Ang2/VEGF-A CrossMab, failed in the phase 2 McCAVE trial (NCT02141295) of colorectal carcinoma and was discontinued [105].

### 3.2. IgG-Modified Bispecific Antibodies, Bivalent or Multivalent

“Two-halves” IgG-like formats are broadly used and very successful, however, additional structural variation offers important advantages in manufacturing and therapeutic application, so that IgG-modified formats are the dominant class today (Figure 2a). Most popular is the Fab-scFv-Fc format, that introduces a second specificity by substituting one Fab region of an IgG mAb with a synthetic scFv (scFv-Fab-Fc, Figure 1d and Figure 2b). A prominent example is the biparatopic anti-HER2/HER2 antibody Zanidatamab (ZW25), based on the Azymetric platform, in which FcγR-binding has been abrogated through “Fc-silencing” mutations to reduce ADCC and side effects from normal tissues [106]. Zanidatamab was granted FDA breakthrough therapy status for advanced biliary tract cancers in late 2020, while it is also evaluated for other HER2-positive advanced solid tumors either alone, or in combination with other treatments (NCT02892123, NCT03929666, NCT04224272, NCT04276493) [107,108]. Further clinical grade mono-scFv-substituted constructs are being based on Xencor’s and Amgen’s heterodimeric platform XmAb, which utilizes engineered Fc isoelectric point differences and amino acid substitutions to achieve heterodimer yields over 95% [109]: the anti-PD-1/CTLA-4 XmAb717 (NCT03517488) and anti-CTLA4/LAG-3 XmAb841 (NCT03849469) for advanced solid tumors, the anti-SSTR2/CD3 XmAb18087 (aka Tidutamab, NCT04590781) for advanced small-cell lung cancer (SCLC) and Merkel cell carcinoma, and the anti-CD20/CD3 XmAb13676 (aka Plamotamab, NCT02924402) for r/r B-NHL. Other similar platforms with clinical grade constructs are Wuhan YZY Biopharma’s YBODY and Glenmark’s BEAT (see Appendix A) [110,111]. On the other hand, substitution of both Fab-regions with scFv gives rise to the half-life extended (HLE) BiTE (Figure 1d, Table 1) and bivalent dual-affinity re-targeting proteins (bi. DART)-Fc (Figure 1d, Table 1), which are also very popular, because they combine the advantages of fragment-based bispecifics with the longer half-life of Fc-based constructs [112,113]. Contemporary HLE BiTEs (Amgen) are Fc-based and have, with dosing every 4–5 days, an efficacy similar to that of daily administered canonical BiTEs, but better than earlier HSA-based HLE BiTE [113]. AMG 673, targeting CD33/CD3, was the first HLE BiTE to enter clinical testing in 2017 for r/r AML (NCT03224819), showing promising efficacy and acceptable toxicity [114]. Currently, at least 8 different HLE BiTEs are in phase 1 trials for hematological and solid cancers (see Appendix A). On the other hand, Fc-based DARTs (MacroGenics) are also dosed in extended intervals, weekly or 3-weekly, in clinical trials since 2014, when the anti-gpA33/CD3 MGD007 entered phase 1 study for advanced colorectal cancer, first alone (NCT02248805), and later in combination with the PD-1 inhibitor MGA012 (NCT03531632) [112]. Another bivalent DART-Fc, the anti-CD19/CD3 MGD011, showed also preclinical potency and favorable pharmacokinetics allowing weekly dosing [115], but was suspended due to neurotoxicity (NCT02454270). Alternatively, the Fab region of IgG can be substituted with a single domain, i.e., VHH (Fab-VH-Fc, Figure 1d), which is more stable under denaturing conditions and less likely to aggregate when multiple fragments are fused together [116,117]. First constructs in the Fc-silenced Fab-VH-Fc format by the proprietary platform UniAbs (Teneobio) entered clinical testing recently, with anti-CD19/CD3 TNB-486 targeting B-NHL (NCT04594642) and anti-PSMA/CD3 TNB-585 targeting prostate cancer (NCT04740034). In preclinical evaluation, both antibodies displayed half-lives similar to IgG antibodies and encouraging efficacy with reduced cytokine release symptoms [118,119].

An important advantage and reason for the upsurge of IgG-modified formats currently (Figure 2a), is the ability to increase the number of binding sites, i.e., valency for the TAA, which translates to increased avidity, enhanced specificity, and the ability of these constructs to compete with monospecific naturally bivalent antibodies (shaded part of Figure 1d, and highlighted in Figure 2b). In fact, multivalent (≥trivalent) formats predominate in clinical trials currently (Figure 2b). The simplest way to produce a multivalent antibody is by chemical cross-linking of two different IgG antibodies (IgG–IgG, Figure 1d and Table 1). Very early trials used such IgG-IgG anti-CD3/CD20 antibodies to “arm” autologous T-cells ex vivo against neoplastic B-cells (e.g., NCT00244946, see Appendix A), while recent studies have expanded this principle to EGFR- or HER2-positive solid tumors using anti-CD3/EGFR or anti-CD3/HER2 antibodies, alone or with immune checkpoint inhibitors (e.g., NCT01420874 and NCT03406858, see Appendix A). Today, with increasing use of genetic engineering, most multivalent antibodies are generated by adding antigen-binding elements, either as (multiple) substitutes for one or more Fab arms, or as appendages to any basic bivalent format.

Fab appendages are a relatively simple and popular way to achieve the aforementioned. Celgene’s CC-93269 combines 2 anti-BCMA with 1 anti-CD3 Fab (Fab3-Fc, Figure 1d and Table 1), and has showed promising preliminary efficacy results in a phase 1 trial of r/r MM (NCT03486067) [120]. Besides, trivalent 2:1 and tetravalent 2:2 CrossMabs contain additional Fab or a CrossFab fused to either the C- or N-terminus of the heavy or light chains [121]. The clinically most advanced CrossMab 2:1 (Figure 2 and Table 1) is the anti-CD20/CD3 Glofitamab, where a CrossFab is inserted between the Fc region and one of the Fab fragments [122]. It is currently entering phase 2, based on a half-life of 6–11 days, similar to Mosunetuzumab, and an overall response rate (ORR) of 66%, with 57% complete response in r/r B-NHL patients who received the recommended phase 2 dose (NCT03075696, NCT04703686) [123,124]. Another CrossMab 2:1 is Cibisatamab, an anti-CEA/CD3 antibody for treatment of solid tumors (see Appendix A). CrossMab 2:2 (Figure 1d and Table 1) is represented by RO6874813, where two anti-FAP CrossFabs are fused to the C-terminus of Drozitumab, an anti-DR5 antibody [125]. FIT-Ig (Fabs-in-tandem, Figure 1d and Table 1) is another Fab-bisubstituted format, in which the light chains of the additional Fabs are fused in tandem with the heavy chains of the core IgG construct (Figure 1d) [126]. Three FIT-Ig antibodies are currently in phase 1/2 clinical trials (see Appendix A).

Even more frequent are scFv-appended formats (Figure 1d). The XmAb AMG 509 features an additional anti-CD3 scFv alongside two identical anti-STEAP1 Fab regions (Fab2-scFv-Fc, Figure 1d and Table 1), showed 50-fold more potent lysis of prostate cancer cells in vitro than XmAb with a single anti-STEAP1 domain [127], and its safety and efficacy in humans is being evaluated in an ongoing phase 1 trial (NCT04221542). Predominant are formats with two additional scFv, which can be attached to various positions on the heavy or light chains. Akeso’s TETRABODY AK104 (anti-PD-1/CTLA-4) is appended at the C-terminus of the heavy chain (IgG(H)-scFv2, Figure 1d and Table 1) and has currently several active clinical trials for various solid tumors (see Appendix A). In a phase 1b/2 study for untreated patients with inoperable gastric or gastroesophageal cancer, it showed an ORR of 60%, a DCR of 93% (n = 7/15) in combination with chemotherapy, and acceptable safety [128]. The scFv fragments can also be appended to the N-terminus of the heavy chains (scFv2-(H)IgG, Figure 1d and Table 1), as in the case of LY3164530 from Eli Lilly [129], an anti-EGFR/cMet antibody that was suspended due to significant toxicity in the phase 1 trial (NCT02221882) [130]. One example of scFv appended on the C-terminus of the light chains (IgG(L)-scFv2, Figure 1d and Table 1) is Nivatrotamab (anti-GD2/CD3) [131], which is based on the GD2-antibody Naxitamab and is being currently assessed in phase 1/2 trials for various solid tumors and specifically metastatic SCLC (NCT03860207, NCT04750239), while constructs appended at the N- terminus of the light chains are in preclinical stage only [4]. The addition of two variable domains pairs (DVD-Ig, Figure 1d and Table 1) creates 4 binding sites and was originally developed to overcome the chain mispairing problem (see Section 2) [55]. Anti-VEGF/DLL4 Dilpacimab (previously ABT-165) is one such antibody. After encouraging efficacy and safety outcomes in preclinical models [132], it advanced and completed phase 1 testing for various advanced solid tumors (NCT01946074), but the phase 2 study for colon cancer in combination with FOLFIRI is currently on hold (NCT03368859).

Increasingly popular are variable domains as multiple substitutions for one or more Fab arms. Aptevo’s ADAPTIR antibodies contain 2 scFv pairs joined with a silenced IgG1 Fc (each linked to the N- or C-terminus, scFv2-Fc-scFv2, Figure 1d and Table 1) [133]. In preclinical studies, the anti-PSMA/CD3 construct APVO414 (aka ES414, MOR209) displayed reduced CRS and a prolonged half-life compared to antibody fragments, allowing for the consideration of weekly doses [133]. However, interim data from the phase 1 trial argue for continuous IV administration, because this decreases formation of anti-drug antibodies [134]. Another similar ADAPTIR antibody is APVO436, directed against CD3/CD123 in phase 1 testing against AML and r/r high-risk MDS (NCT03647800). Alternatively, 2 DARTs attached to an Fc region (tetra-DART-Fc, Figure 1d and Table 1) result in bivalence for each target [135]. Such constructs in clinical trials currently are the anti-PD-1/CTLA-4 MGD019 (NCT03761017) [136], and the anti-PD1/LAG-3 MGD013 (Tebotelimab) in various advanced solid tumors (NCT03219268), both with acceptable safety profiles and very promising activity in the respective phase 1 results published in 2020 [137,138]. A functionally similar result can be achieved by attaching 2 pairs of nanobodies to Fc in a symmetric manner (VHH4-Fc, Figure 1d and Table 1), as is the case for both INBRX-105 (ES101 in China) and KN046. INBRX-105 is an Fc-silenced anti-PD-L1/4-1BB antibody, and KN046, an anti-PD-L1/CTLA-4 antibody, both in several early trials for various metastatic solid tumors (see Appendix A) [139,140]. Substitution of one Fab arm with one pair of identical nanobodies (Fab-VH2-Fc, Figure 1d and Table 1) is the format of the trivalent bispecific IBI322 targeting CD47(Fab)/PD-L1(VH2) [141], for which clinical trials were very recently launched in the USA and China (NCT04338659, NCT04328831). TNB-383B is a similar anti-CD3(Fab)/BCMA(VH2) construct by TeneoBio based on the UniAbs platform, which has demonstrated very low CRS rate, solely grade 1–2 occurring in only 21% of the patients (n = 38) enrolled in a phase 1 trial [142,143].

A completely different strategy has been implemented by F-star, whose mAb^2^ technology introduces two new binding sites directly in the Fc region (IgG(H)-Fcab2, Figure 1d and Table 1), while simultaneously maintaining Fc binding to the Fcγ and FcRn receptors [144]. In a phase 1 trial of the FS118 directed against PD-L1(Fab)/LAG-3(Fcab2), disease stabilization was observed in a subset of patients who had acquired resistance to PD-(L)1 therapy, but not in patients with primary resistance (NCT03440437) [145,146]. Treatment-related adverse events were more frequently of grades 1–2. Other antibodies of the same format are also under phase 1 evaluation in advanced malignancies (FS120 directed against CD137(Fab)/OX40(Fcab2), NCT04648202, and FS222 directed against anti-PD-L1(Fab)/CD137(Fcab2), NCT04740424).

### 3.3. Fragment-Based Bispecific Antibodies (Fc-Free), Bivalent or Multivalent

Among fragment-based, Fc-free bispecifics, the tandem scFv-based BiTEs (Figure 1e and Table 1) predominate (Figure 2b). The anti-CD19/CD3 BiTE blinatumomab was a first-in-class construct and remains the only approved bispecific in oncology, used for r/r or minimal residual-disease (MRD)-positive B-ALL [147,148]. A feasibility study to evaluate outpatient treatment of MRD-positive patients is about to start recruiting in May 2021 (NCT04506086). Besides, the anti-BCMA/CD3 BiTE AMG-420, formerly known as BI 836909, has shown promising activity in r/r MM, including MRD-negative complete responses (NCT02514239) [149], with an ongoing expansion study (NCT03836053), while the anti-CD33/CD3 BiTE AMG-330 is currently in phase 1 testing (NCT02520427) [150]. Another elegant platform is GEMoaB’s “affinity-tailored adaptors for T-cells” (ATAC) fully humanized tandem scFv platform, which employs high binding affinity to TAA and lower affinity to CD3, thus preventing T-cell auto-activation in preclinical models [151]. Two ATACs are currently in clinical testing with no published results yet: the anti-CD33/CD3 GEM333 for r/r AML (NCT03516760), and the anti-PSCA/CD3 GEM3PSCA for various advanced solid tumors (NCT03927573). At the same time, MM-111, consisting of human anti-HER2 and anti-HER3 scFv linked by modified HSA [152], is currently held in reserve by Elevation Oncology, which acquired it from Merrimack in 2019 after a negative phase 2 trial in HER2-positive gastric and esophageal tumors (NCT01774851), but currently pushes development of seribantumab (MM-121, a monospecific HER3-monocloncal antibody also acquired from Merrimack at the same time) for tumors harboring NRG1-fusion.

On the other hand, the diabody-based DART format (Figure 1e and Table 1) was developed with the goal of improving the geometry of bispecific interactions. It has a very compact and stable structure, due to cross-linking of V_H_ and V_L_ chains of different specificities as well as stabilization by disulfide bridges, and could indeed demonstrate stronger redirected T-cell-mediated killing of CD19-expressing malignant B-cells in vitro [153]. DART development is currently focused on the half-life extended bivalent and tetravalent DART-Fc (presented in Section 3.2), so that the only canonical DART in phase 1/2 testing today is the anti-CD123/CD3 Flotetuzumab (MGD006) for r/r AML (NCT02152956), with ORR of 24%–30% [154]. Common adverse events were infusion-related reactions: CRS (96%), nausea (26%), and cytopenia [154].

The main strategy to increase valency in fragment-based bispecifics are tandem diabodies (TandAb, Figure 1e and Table 1), i.e., are tetravalent bispecific molecules resulting from the non-covalent homodimerization of two single-chain diabodies [155,156]. Currently, the only TandAb in active clinical testing is AFM13, an NK-cell redirecting anti-CD30/CD16A TandAb for the treatment of lymphomas, while the anti-CD19/CD3 AFM11was suspended due to side effects (NCT02106091). The safety profile of AFM13 was acceptable with a frequency of grade ≥3 adverse events at 29% in the phase 1 trial (NCT01221571). The mean terminal half-life 9 to 19 h at a size of 104 kDa [155,157] permits weekly dosing in the current phase 2 trial (NCT04101331), after the continuous 5-day-long infusions weekly in a previous phase 2 trial (NCT02321592) had prohibited sufficient recruitment [158]. Combination of AFM113 with the PD-1 inhibitor pembrolizumab appears to increase efficacy and warrants further evaluation [159], while a trial assessing administration of AFM113 with autologous NK cells after fludarabine/cyclophosphamide conditioning is ongoing (NCT04074746).

An earlier Fab-based (Table 1) anti-EGFR/anti-FcγRI bispecific construct did not show clinically relevant activity against advanced solid tumors [160]. Another tandem F(ab′)2 fragment against CD28 and a melanoma-associated proteoglycan synthesized by chemical hybridization in the Tübingen University was also abandoned after phase 1 testing (NCT00204594) [161]. Bispecific nanobodies lacking an Fc region have not found their way to clinical trials, possibly because of poor pharmacokinetics due to their very small size.

### 3.4. Antibody-Based Bispecific Fusion Proteins

Antibodies can acquire additional functionality through conjugation with additional moieties. One main trend is inhibition of TGFβ signaling using TGFβ-traps (Figure 1f and Table 1), with the most advanced being Merck’s bintrafusp alfa (aka M7824, overall >40 trials, see Appendix A), a fusion protein that contains the extracellular domain of TGFBR2 fused to an anti-PD-L1 IgG1 [162]. It has shown promising activity in phase 1 testing for several solid cancers [163], but did not meet the efficacy threshold in a phase 2 trial for advanced biliary cancer (NCT03833661) [164]. Besides, GS-1423 is an anti-CD73 IgG, whose Fc region is linked with two TGFβ-traps in the form of the TGFβRII extracellular domain [59], and is currently undergoing testing against various solid tumors (NCT03954704), while BCA101, an EGFR-specific antibody with two TGFβ-trap moieties, is tested against solid tumors in a phase 1 trial (NCT04429542). Additional TGFβ-trap fusion proteins are in preclinical development, e.g., TST005 with two stable TGFβ-traps, each linked on the heavy chain C-terminus of an anti-PD-L1 IgG [165]. Less frequently, other fused moieties are employed, for example, AMG 256 is a PD-1 inhibitor with an Fc-linked IL-21 receptor agonist, currently tested in various solid tumors (NCT04362748) [166], while IMM0306, an anti-CD20 IgG linked to SIRPα [167], is tested for r/r B-NHL (NCT04746131).

Undoubtedly, the most significant bispecific fusion protein format currently are Immunocore’s immune-mobilizing monoclonal T-cell receptors against cancer (“ImmTACs”, Figure 1e and Table 1). These are 75 Da molecules that combine a high-affinity, HLA-A*0201-restricted TCR specific for TAA peptides presented on MHC (p.MHC) with a humanized scFv against CD3, and redirect T-cells against intracellular antigens [58]. Most advanced is IMCgp100, aka tebentafusp, which recognizes a gp100-derived peptide, and recently demonstrated prolongation of overall survival as first-line monotherapy for patients with metastatic uveal melanoma in a randomized phase 3 trial (NCT01211262) [168]. Tebentafusp is expected to achieve regulatory approval soon and is already accessible within an international compassionate use program. Two other ImmTACs are in phase 2 testing, namely IMC-C103C for MAGE-A4-derived peptides [169] (NCT03973333), and IMC-F106C for “preferentially expressed antigen in melanoma” (PRAME [170])-positive tumors (NCT04262466), both exploiting the HLA-A2 background present in approximately 50% of Caucasian patients [171].

## 4. Trispecific and Other Multispecific Antibodies

Further increase of antibody specificity can potentially result in even higher efficacy compared to mono- and bispecific antibodies. Indeed, several multispecifics have already shown promising data and are entering clinical testing [172]. The main application field remains oncology, with generally at least one of the three specificities intended to redirect T- or NK cells, and the other two targeting TAA. Some of them are not stricto sensu trispecific antibodies: for example, the scFv-based TriKEs (Figure 1e and Table 1) bind one TAA and CD16, while their third moiety is IL-15, which activates the recruited NK cells through cross-linking of their IL-15 receptors without involvement of an antibody-antigen interaction (Figure 1e) [173]. Nevertheless, since they are antibody molecules working by triple interaction, they will be considered here, as well.

Currently, there are several such trispecifics in clinical trials, most of which are immune cell engaging. The TriKE GTB-3550, developed by GT Biopharma, targets CD33 and is currently tested in a phase 1/2 study for r/r high-risk myeloid malignancies (NCT03214666). Of note, GT Biopharma is also developing two TriKEs against solid tumors: GTB-4550 targeting PD-1, and GTB-5550 targeting B7H3 [174]. Additional TriKEs are in preclinical stage, such as 161,533 targeting CD33 [175], or a slightly different construct targeting B7-H3 and using an sdAb to bind CD16 [176]. Another related variant are trispecific “NK-cell engagers” (NKCE), which recruit NK cells by using scFv or Fab against TAA, NKp46, and CD16 [177]. Another trispecific developed by the Swiss company Numab, the scFv-based NM21-1480 (scFv3, Figure 1e and Table 1) is not a classical T-/NK cell engager, but rather activates immune cells by simultaneous binding of PD-L1, 4-1BB, and HSA (the latter to prolong the half-life) and is currently in phase 1/2 testing against advanced cancers (NCT04442126).

Four other constructs in phase 1/2 trials are diabody-based “Trispecific T-Cell activation Constructs” (TriTACs, Figure 1e and Table 1) developed by Harpoon Therapeutics, which consist of a nanobody targeting the respective TAA, joined with an anti-CD3 scFv, and a nanobody against HSA (serving to prolong the half-life). HPN 328 targets DLL-3 and is tested against small-cell lung cancer (NCT04471727), HPN217 targets BCMA and is tested in r/r MM (NCT04184050), HPN424 targets PSMA and is tested in advanced prostate cancer (NCT03577028), while HPN536 targets mesothelin (MSLN) and is tested in various advanced solid tumors, including ovarian or pancreatic cancer, and malignant pleural mesothelioma (MPM, NCT03872206). Another trispecific T-cell engaging antibody is the IgG-like SAR442257, of which one Fab fragment targets CD38, while the second carries a cross-over dual variable (CODV) region specific for CD3 and CD28 [178]. A first-in-human study of SAR442257 in r/r MM and non-Hodgkin’s lymphoma is ongoing (NCT04401020). Other TriTACs are in Harpoon’s pipeline, e.g., G3, G4, and G8 targeting FLT3 [179], while first preclinical data also suggest synergy with PD-(L)1 inhibitors [180]. A novel platform very similar to the TriTACs are designed ankyrin repeat proteins (DARPins, Figure 1e and Table 1) developed by molecular partners in cooperation with Amgen. These are antibody-mimetic 62 kDa polypeptide chains containing 4 ankyrin domains with specificity against HSA (2×), and either two TAA (1× each), or one TAA (1×) and one T-cell antigen (1×), and which can be easily produced in bacteria [181]. MP0250 targeting VEGF/HGF is in phase 2 testing for r/r MM (NCT03136653), MP0310 (aka AMG 506) directed against the fibroblast activation protein [182] (FAP)/4-1BB, and MP0274 targeting two different HER2 epitopes are in phase 1 testing (NCT04049903, NCT03084926), while MP0317 targeting FAP/CD40, and various T-cell redirecting DARPins (CD3-engaging) are expected to follow soon.

Several other trispecific platforms also exist, either bulkier, such as the older Dock-and-Lock (DNL, Figure 1g) method based on the dimerization domain of cAMP-dependent protein kinase A and the anchoring domain of an A-kinase anchoring protein [183,184], or the newer, more compact VHH-based trispecific constructs, which are still at the preclinical stage [185]. In addition, IgG/scFV-based tetraspecifics (scFv6-IgG, Figure 1d and Table 1), have recently crossed the border to clinical testing. GNC-038 and GNC-039 (for “guidance, navigation and control”) of Bailey Pharmaceuticals and its subsidiary Systimmune, employ a novel octavalent format to engage and activate T-cells by binding CD3 and 4-1BB, while simultaneously also inhibiting PD-L1 on tumor cells and binding a TAA, namely CD19 and EGFRvIII on r/r NHL and various solid tumors, respectively (NCT04606433, NCT04794972). A third similar construct, GNC-035 targeted against the oncoembryonic antigen ROR1 expressed by a variety of human cancers [186], is expected to also enter clinical testing soon. It remains to be shown whether multispecific constructs under development will surpass, in effectiveness, bispecifics and their combinations [172].

## 5. Payload Delivery Using Multispecific Constructs

The spectacular enhancement of monospecific antibody efficacy through conjugation with toxins, as exemplified by trastuzumab emtansine [187], trastuzumab deruxtecan [188], and brentuximab vedotin [189], has sparked great interest for multispecific antibody drug conjugates (ADC, Figure 1g and Table 1). In addition, antibody-fragment-drug conjugates (FDC, Figure 1g and Table 1) have emerged more recently as “next-generation ADC” particularly suited for treatment of solid tumors, building on the improved tissue penetration of smaller molecules, and reduced payload exposure of normal tissues due to lack of Fc-based interactions and the shorter half-lives [190]. Therefore, both Fc-based and Fc-free multispecific formats are increasingly used for payload delivery, aiming for more precise targeting of tumor cells and enhanced potency due to engagement of additional epitopes.

Currently, a few different bispecific antibodies for payload delivery are in clinical testing. TargomiR EDVs coated with IgG-IgG (EDV, Figure 1g, anchored via *S. Typhimurium* O-antigen on the minicell surface, and targeting EGFR) and loaded with a miR-16-based miRNA-mimic were used against MPM and NSCLC in a phase 1 trial. Partial responses were noted in 1/22 (5%) patients, and stable disease in 15 (68%), while dose-limiting toxicities were infusion reactions, coronary ischemia, anaphylaxis, cardiomyopathy and non-cardiac pain (NCT02369198) [191]. Another phase 1 study is testing an EGFR-directed mitoxantrone-loaded EDV (EEDVSMit) in children with r/r solid tumors (NCT02687386). A third clinical trial utilized EGFR(V)-EDV-Dox in patients with recurrent glioblastoma (NCT02766699), after it had shown significant prolongation of overall survival in two orthotopic human neuroblastoma xenograft models [192]. Eight of 14 patients completed one cycle, four more than one cycle of therapy; median PFS was 1.6 months, 2/14 patients had a PFS >6 months, and median OS was 9.7 months. Side effects like fever, nausea or chills occurred but were manageable [193].

On the other hand, M1231 is a bispecific Fab/scFv ADC (Fc-based) directed against EGFR/MUC-1 undergoing phase 1 testing for metastatic solid tumors (NCT0469584) [194], while ZW49 is an ADC of the HER2-bispecific antibody Zanidatamab conjugated with a novel auristatin payload using a cleavable linker, which is currently evaluated in advanced HER2-expressing malignancies in a global phase 1 trial [195]. DT2219ARL, an immunotoxin composed of CD19 and CD22 scFv linked to diphtheria toxin, is the only FDC developed for hematologic malignancies, and has shown activity in phase 1/2 trials of r/r B-cell leukemia or lymphoma (NCT00889408, NCT02370160) [196]. In 12 treated subjects, 4 achieved stable disease and 1 had partial response [197].

Bispecific antibodies can also be used for pretargeted radioimmunotherapy (pRAIT). TF2, a bispecific tri-Fab antibody, targeting HSG (1×, “histamine-succinyl-glycine”, a unique synthetic hapten, which has been incorporated in several small peptides that can be labeled with a wide range of radionuclides [198]) and CEA (2×), is produced with the DNL method (DNL, Figure 1g). For pRAIT, first, the bispecific antibody TF2 is infused, followed by infusion of the radiolabeled IMP-288 that binds to the TF2 on the cancer cells. “Pretargeting” should lead to a stronger radiation signal, while reducing the dose and toxicity of normal tissue [199]. Efficacy of TF2 in combination with Lu-177-labeled IMP-288 is evaluated in two clinical trials for colorectal or lung cancer (NCT00860860, NCT01221675). Best results were achieved with a shorter pretargeting delay (24 h) and higher dose (480 nmol/m^2^), but outcome has been mediocre: 4 patients were stable at 4 weeks, but progressive at 3 months, while 6 patients were progressive already after 4 weeks [200]. For pRAIT of medullary thyroid cancer, the chemically cross-linked human/murine bispecific anti-CEA/anti-DTPA antibody hMN-14 × m734 [201] is tested in combination with di-DTPA-131I (NCT00467506). Besides, pretargeting with bispecific antibodies is also under investigation for imaging, e.g., IMP 205 × m734 is infused, followed by Indium-labeled IMP-205 for imaging of colorectal cancer (NCT00185081). An interesting construct still at preclinical stage is the bispecific MaAbNA (multivalent antibody comprised of nanobody and affibody moieties), built by an anti-EGFR sdAb and two anti-HER2 affibodies, and conjugated with adriamycin [202]. Affibodies are engineered, small (6.5 kDa), high affinity antibody-mimetics, which are based on the IgG binding domain of staphylococcal protein A [203,204].

## 6. Challenges and Perspectives

While multispecific antibodies are a very dynamic and promising field, it still faces some major challenges. The most serious difficulty is certainly the availability of suitable target antigens for the majority of cancers. While some differentiation antigens expressed in the hematologic lineages are dispensable, for example CD19, CD33, BCMA, and therefore suitable for attack with exquisitely potent therapeutics, as are multispecifics, there is no corresponding target that can be safely eliminated in epithelial tumors [205]. In addition, it is estimated that only approximately 10% of human proteins are on the cell surface and thus accessible to classical antibodies [206,207]. One important development in this respect was the successful targeting of G12V and Q61H/L/R *KRAS* mutant cells using single-chain diabodies specific for the respective HLA-A1/3-neopeptide complex and CD3, which paves the way for therapeutic exploitation of intracellular neoantigens with multispecific antibodies [208]. Other approaches to target neoantigens (“MANAbodies” for “mutation-associated neoantigen-directed antibodies”) or intracellular TAA, utilize either ImmTACs (described in Section 3.4 [58]), or suitable TCR-mimic mAb in the BiTE format [209]. An alternative approach to overcome extracellular target scarcity are combinatorial formats in preclinical stage; for example, hemibodies: two scFv, each directed against a different TAA and linked with either the V_H_ or the V_L_ domain of a composite anti-CD3 Fv, become active and engage T-cells when they join after encountering both TAAs on the same tumor cell surface [210].

Beyond the problem of suitable antigens, several other reasons can result in failure of a promising construct, as the several terminated clinical trials show (see Appendix A). One main problem is the lack of efficacy, as with TrioMabs other than catumaxomab or with vanucizumab (see Section 3.1), for which the target antigens chosen and/or effect strength were not sufficient for clinically relevant benefit. Besides, toxicity can also be limiting for some highly potent constructs, such as the anti-CD19/CD3 DART-Fc MGD011 (see Section 3.2). Additionally, since the development costs of multispecifics are considerable, slow accrual in a clinical trial, for example, due to complex logistics combined with a very narrow patient population, can force the investigators to cancel the effort when the sponsor or company producing the antibody run out of money (e.g., in the case of FBTA05 administered with donor lymphocyte infusions after allogeneic hematopoietic cell transplantation (alloSCT, NCT01138579), or IgG-IgGs administered with autologous T-cells for EGFR- or HER2-expressing solid tumors (NCT02470559 and NCT02521090)).

Another important issue for immune-cell redirecting antibodies, such as BiTEs, is that their mode of action is very similar to that of cell therapies, for example, chimeric-antigen-receptor T (CAR-T) cells directed against the same antigen. One main advantage of antibodies is broad, “off-the-shelf” availability, in contrast to CAR-T cells, which currently require a several weeks-long and very expensive manufacturing process for each individual patient, some of which will not live to receive the cellular product [211]. In addition, multispecific antibodies appear to have better tolerability, since they do not require lymphodepletion prior to infusion, while cytokine-release syndrome (approximately 5% vs. 50%) and neurotoxicity (<10% vs. 12–32%) have generally been less frequent, less severe, and better manageable with the short-acting CD19-specific BiTE blinatumomab than after infusion of the long-persisting CD19-specific CAR-T cells [212]. On the other hand, efficacy appears to be somewhat better with CAR-T cells than blinatumomab, with higher response rates in r/r acute lymphoblastic leukemia (approximately 80% vs. 40%), especially in cases of high tumor burden (e.g., bone marrow blasts >50%), extramedullary disease, and central nervous system involvement [213]. Likewise, in r/r MM BCMA-specific CAR-T cells achieve response rates >90%, while the response rate in the recent phase 1 trial of a BCMA/CD3 bispecific was 33–75% [214,215]. In the r/r follicular lymphoma (FL) and DLBCL, response rates also appear to be higher with Axi-cel (>90% and >70% respectively, FDA-approved for both entities) than with blinatomumab (approximately 80% and 55%, respectively) [216,217]. However, cross-series comparisons may not be accurate due to patient heterogeneity, while treatments are rapidly evolving; for example, the newer CD20-specific bispecific antibody odronextamab (REGN1979) could recently also achieve a response rate >90% in r/r FL [218]. Overall, multispecifics and CAR-T therapies both represent major breakthroughs in cancer therapy and will probably become complementary in the future, one example being BiTE-secreting CAR-T cells, which circumvent escape due to antigen loss without detectable toxicity [219]. Of note, bispecific antibodies can also be used to optimize the outcome of patients after alloSCT [220], which is still the mainstay of treatment for many r/r or high-risk myeloid and lymphatic malignancies [221,222].

Along the same lines, multispecific constructs will probably need to be combined with other immunotherapies, for example, immune checkpoint inhibitors [223] and oncolytic viruses [224], in order to overcome the immune dysregulation of various cancers [225,226,227,228,229] and maximize clinical benefit. Novel multispecific constructs have already incorporated some essential synergistic effects, for example, “checkpoint-inhibitory T-cell engagers” (CiTEs) are BiTEs equipped with an additional moiety blocking the PD-1/PD-L1 axis or another immune checkpoint [230], while “simultaneous multiple interaction T-cell engagers” (SMITEs) are the combined administration of different BiTEs in order to target multiple TAAs and/or turn tumor-cell inhibitory (e.g., from PD-L1) into T-cell costimulatory (e.g., via CD28) signals [231,232]. The development of such multimodal constructs directed against intracellular neoantigens could become a turning point in the treatment of solid cancers.

## 7. Conclusions

Multispecific antibodies are a rapidly growing field with huge therapeutic potential. Accumulating preclinical and early clinical data provide ample evidence on how targeting multiple tumor cell antigens and additional recruitment of effector lymphocytes increase therapeutic efficacy and could further improve clinical outcomes. Considering the fact that introduction of PD-(L)1 inhibitors a few years ago has already facilitated 5-year survival rates of 30–50% for patients with per se highly lethal neoplasms, such as metastatic melanoma and non-small-cell lung carcinoma [233,234], the upcoming complementation of current strategies with “next-generation” immunotherapeutics, such as multispecific antibodies and cell therapies, offers a justified hope for the cure of some advanced cancers in the near future.

## Figures and Tables

**Figure 1 ijms-22-05632-f001:**
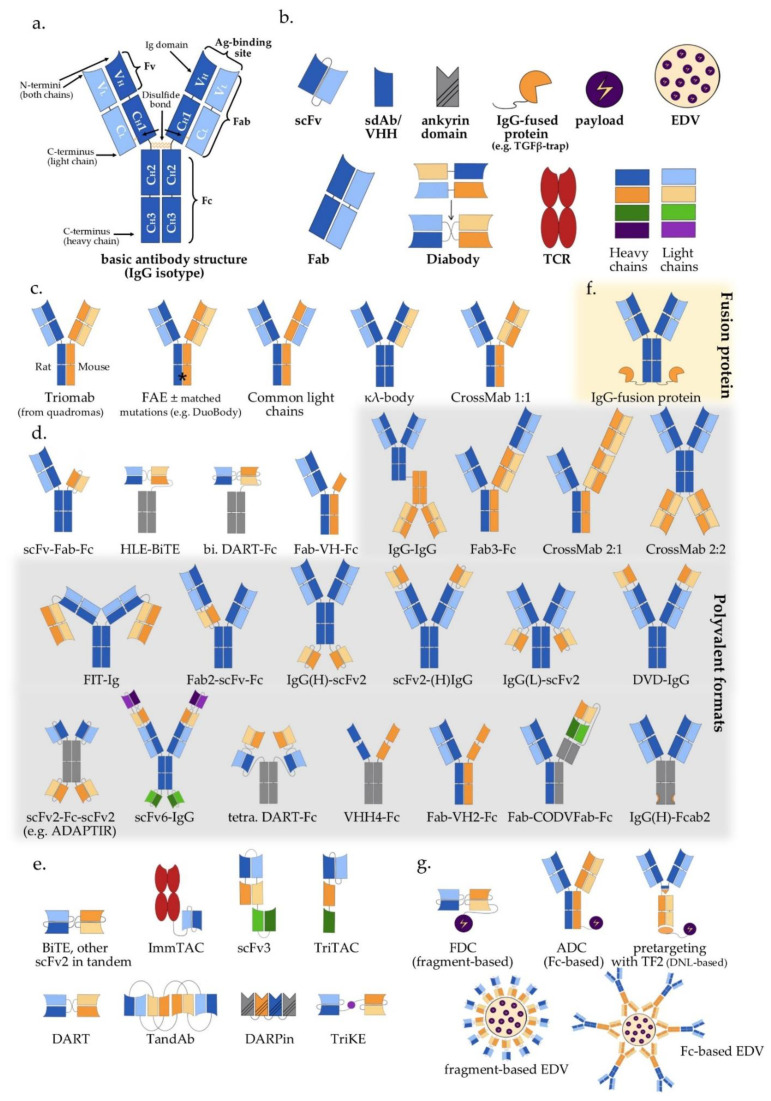
Multispecific antibody formats in clinical trials: (**a**) Basic IgG structure; (**b**) Main antibody components (detailed in Section 2 of the main text); (**c**) IgG-like “two-halves” bispecific formats (following the order of Section 3.1 and Table 1); (**d**) IgG-modified (appended and/or substituted, Fc-based) bispecific and multispecific antibodies (following the order of Section 3.2 and Section 4, as well as Table 1); (**e**) Fragment-based (Fc-free) bispecific and multispecific antibodies (following the order of Section 3.3 and Section 4, as well as Table 1); (**f**) IgG fusion protein; (**g**) Bi-/multispecific constructs for payload delivery (following the order of Section 5 and Table 1). For the abbreviations, please see the respective section in the main text.

**Figure 2 ijms-22-05632-f002:**
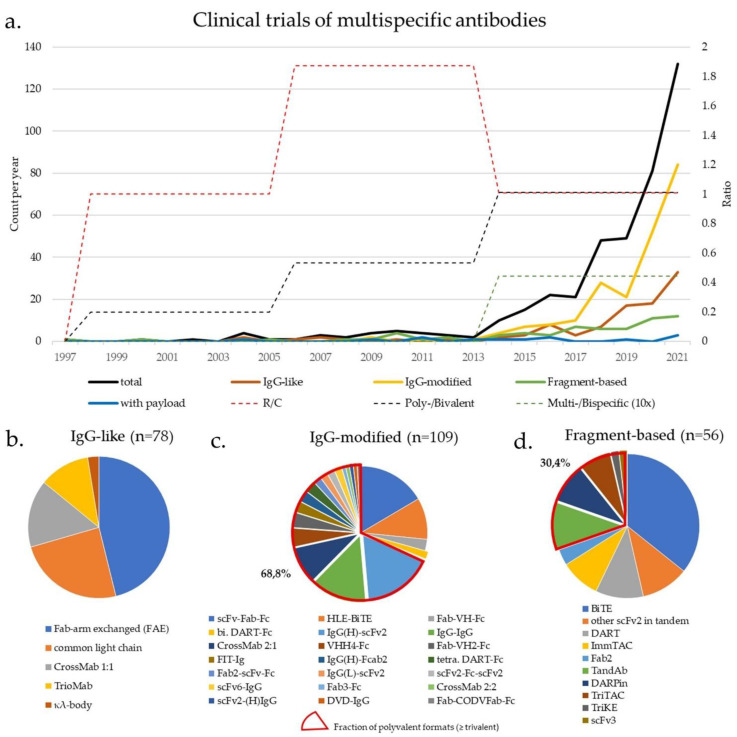
Landscape of clinical trials using multispecific antibodies in oncology: (**a**) Longitudinal changes in the cumulative number of initiated clinical trials since 1997 for each of the main categories according to Table 1, as well as in the ratios of constructs with “redirecting”/”classical” mode of action, and ≥trispecific/bispecific (×10), or ≥trivalent/bivalent formats in 1997–2005, 2006–2013, and 2014–2021. The search strategy is described in Section 3. For 2021, only studies until April were considered, and their number was corrected for the smaller time duration (4/12 months); (**b**–**d**) Distribution of clinical trials among individual formats, according to the classification in Table 1. Fab-arm-exchanged constructs dominate among IgG-like formats, IgG(H)-scFv2 or scFv-Fab-Fc, and multivalent constructs dominate among IgG-modified formats, while BiTEs dominate among fragment-based formats. The numbers are shown in Table 1, the structural details in Figure 1, while the complete list of trials is given in the Appendix A (*n* = 324, of which 309 with reported molecular formats). Fusion proteins were excluded from this analysis.

**Table 1 ijms-22-05632-t001:** Multispecific antibody formats in clinical trials by structure, mode of action and start year of first study.

	Class	Specificity/ Valence	Action (C/R)	Format	No. of Clinical Trials	Comment	First Clinical Trial
**Fc-based**	**IgG-like** (Section 3.1) (Figure 1c)	2/2	R	TrioMab	9	“two-halves” formats	2004
		2/2	C/R	FAE	36		2010
		2/2	C/R	common light chain	19		2014
		2/2	C	κλ-body	2		2019
		2/2	C/R	CrossMab 1:1	12		2012
	**IgG-modified** (Section 3.2 & Section 4) (Figure 1d)	2/2	C/R	scFv-Fab-Fc	18	scFv-monosubstituted	2016
		2/2	R	HLE-BiTE	11	scFv-bisubstituted	2015
		2/2	R	DART-Fc	2	Db-bisubstituted	2014
		2/2	R	Fab-VH-Fc **	3	V_H_-monosubstituted	2021
		2/4	R	IgG-IgG	15	IgG-IgG	2004
		2/3	R	Fab3-Fc	1	Fab-appended	2018
		2/3	R	CrossMab 2:1	10		2014
		2/4	C	CrossMab 2:2	1		2015
		2/4	C/R	FIT-Ig (Fabs-in-Tandem)	3		2018
		2/3	R	Fab2-scFv-Fc	2	scFv-appended	2020
		2/4	C/R	IgG(H)-scFv2	19		2017
		2/4	C	scFv2-(H)IgG	1		2014
		2/4	R	IgG(L)-scFv2	2		2019
		2/4	C	DVD-IgG	1	V-appended	2013
		2/4	R	scFv2-Fc-scFv2	2	scFv-multisubstituted	2015
		4/8	R	scFv6-IgG	2		2020
		2/4	C	DART-Fc	3	Db-multisubstituted	2017
		2/4	C	VHH4-Fc	5	V-multi substituted	2019
		2/3	C/R	Fab-VH2-Fc **	4		2019
		3/3	R	Fab-CODVFab-Fc	1		2020
		2/3-4 *	C	IgG-fusion proteins	55	Fusion moiety	2015
		2/4	C	IgG(H)-Fcab2	3	Fc-modified	2018
**Fc-free**	**Fv-based ^$^** (**^$^**Section 3.3 & Section 4) (**^$^** Figure 1e)	2/2	R	BiTE	20	scFv-based	2008
		2/2	C/R	other scFv2 in tandem	6		2005
		2/2 *	R	ImmTAC	5		2015
		3/3 *	R	TriKE	1		2020
		3/3	C	scFv3	1		2020
		2/2	R	DART	6	Db-based	2014
		2/4	R	TandAb	6		2010
		3/3	R	TriTAC	4		2018
		¾ *	C	DARPin	5	Ankyrin-based	2014
	**Fab-based** **^$^**	2/2	C	Fab2	2	Fab-based	1997
**with payload** (Section 5) (Figure 1f)		2/2 #	C	Fc-based ADC/EDV	4	IgG-based	2014
		2/2 #	C	Fc-free FDC/EDV	3	Fragment-based	2013
		2/2-3 (#)	C	±pretargeting (±imaging)	4		2004

Formats are ordered as in Figure 1 and the corresponding sections of the main text; C: classical mode of action; R: immune-cell redirecting; for the explanation of other abbreviations, please see the main text; * one binding site does not rely on typical antigen-antibody interaction (bispecifics described in Section 3.4, while trispecifics in Section 4); ** human VH or VHH; # with payload; ^$^ details in Section 3.3 and Section 4, as well as Figure 1e.

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
