# Peer review of "Principles and Current Clinical Landscape of Multispecific Antibodies against Cancer"

_ijms, 2021, doi:10.3390/ijms22115632_

Round 1

Reviewer 1 Report

The review article entitled Principles and current landscape of multispecific antibodies against cancer by Elshiaty et al is an analysis of the current context of antibodies as cancer immunotherapies.

The review is well-written and very comprehensive with an emphasis on clinical trials identified through the ClinicalTrials.gov site as stated in the review. The review encompasses a description of the different types of antibody constructs and their current status as therapies. This is likely to be a valuable reference document for researchers and clinicians interested in immunotherapy development.  

I have some minor comments.

Additional formatting would help the reader navigate this document. Some sections are long and breaks or subheadings would be helpful

Could the authors add a section on promising but failed antibodies or antibody constructs? Any lessons to be learned from these failures?

Are there any new types of antibody constructs under consideration? What further advantages could be added to the current (already extensive) library of constructs?

Line 718, please change “Priori to infusion” to “Prior to infusion”

Author Response

We would like to thank the reviewer for his encouraging comments. According to his suggestion and in order to facilitate better navigation, we have split some sections, improved some headings, and added to every format mentioned in the text a reference to the exact designation in Figure 1 and Table 1, which now help better maintain an overview.  In addition, the Figures (particularly Figure 2), have been improved to provide additional, more general information for a better overview of the field (i.e. the rations of various characteristics in Fig. 2A, and the fraction of multivalent constructs in Fig. 2B). Furthermore,  we have added a section on promising but failed antibody constructs in section 6, along with some important reasons and lessons to be learned. In section 6, we also detail some important prospects and new construct types not yet in clinical testing: strategies to target intracellular neoantigens (beyond ImmTACs), which is of paramount importance due to the scarcity of suitable surface antigens in solid tumors, and the combination with other immunotherapies, CiTEs and SMITES, which is essential because of the complex immune dysregulation associated with solid cancers (last paragraph of section 6). “Prior to infusion” has now been corrected in line 718 of the original manuscript, thank you.

Reviewer 2 Report

This manuscript surveys the structures and mechanistic principles of multispecific antibodies used in clinical trials against cancer. Therefore, its focus is on the technical development and construction principles of these promising agents rather than therapeutic efficacies. Consequently this manuscript provides a comprehensive picture of a field that often is treated as a side aspect in reports about their application that are bound to center on treatment efficacies against diverse types of cancer.

Author Response

We would like to thank the reviewer for his encouraging comments. Indeed, we dedicate a greater part to principles and technical development, because for most currently available constructs, the (efficacy) results of clinical trials are still pending. Nonetheless, we provide also efficacy data for newer constructs with available clinical results, like tebentafusp, flotuzumab and several others, and also consider the efficacy/potential clinical utility of multispecifics in comparison to cell therapies, which are the most important alternative strategy. In the revised version of our manuscript, we have undertaken several small improvements in order to enhance readability and navigation in the complex material.